# In situ dual doping for constructing efficient $CO_2$-to-methanol electrocatalysts

Pengsong Li[1,2,3,4], Jiahui Bi[1,2,3,4], Jiyuan Liu[1,2,3,4], Qinggong Zhu ●[1,2,3,4 ✉], Chunjun Chen[1,2,3,4], Xiaofu Sun[1,2,3,4], Jianling Zhang[1,2,3,4] & Buxing Han ●[1,2,3,4,5 ✉]

Methanol is a highly desirable product of $CO_2$ electroreduction due to its wide array of industrial applications. However, the development of $CO_2$-to-methanol electrocatalysts with high performance is still challenging. Here we report an operationally simple in situ dual doping strategy to construct efficient $CO_2$-to-methanol electrocatalysts. In particular, when using $Ag,S$-$Cu_2O/Cu$ as electrocatalyst, the methanol Faradaic efficiency (FE) could reach 67.4% with a current density as high as 122.7 mA cm$^{-2}$ in an H-type cell using 1-butyl-3-methylimidazolium tetrafluoroborate/$H_2O$ as the electrolyte, while the current density was below 50 mA cm$^{-2}$ when the FE was greater than 50% over the reported catalysts. Experimental and theoretical studies suggest that the anion S can effectively adjust the electronic structure and morphology of the catalysts in favor of the methanol pathway, whereas the cation Ag suppresses the hydrogen evolution reaction. Their synergistic interactions with host material enhance the selectivity and current density for methanol formation. This work opens a way for designing efficient catalysts for $CO_2$ electroreduction to methanol.

[1] CAS Key Laboratory of Colloid, Interface and Chemical Thermodynamics, Institute of Chemistry, Chinese Academy of Sciences, 100190 Beijing, P. R. China. [2] Beijing National Laboratory for Molecular Sciences, Institute of Chemistry, Chinese Academy of Sciences, 100190 Beijing, P. R. China. [3] CAS Research/Education Center for Excellence in Molecular Sciences, Institute of Chemistry, Chinese Academy of Sciences, 100190 Beijing, P. R. China. [4] University of Chinese Academy of Sciences, 100049 Beijing, P. R. China. [5] Shanghai Key Laboratory of Green Chemistry and Chemical Processes, School of Chemistry and Molecular Engineering, East China Normal University Shanghai, 200062 Shanghai, P. R. China. ✉email: qgzhu@iccas.ac.cn; hanbx@iccas.ac.cn

Electrochemical carbon dioxide reduction reaction (CO$_2$RR) to high-value-added chemicals or fuels, driven by renewable energy sources, is a promising strategy to reduce greenhouse gas accumulation and simultaneously provide an avenue toward the global carbon cycle[1–4]. As a C$_1$ product of CO$_2$RR, methanol possesses the advantages of high-energy density, easy storage, and transportation under ambient conditions, and is also a vital chemical feedstock for plastic, paint, and silicone[5,6]. At present, it is mainly manufactured through fossil-based syngas, and the traditional process emits a large amount of CO$_2$ (about 2.6 ton CO$_2$ / ton methanol)[7]. Although direct electrochemical CO$_2$RR poses the great potential to shift the paradigm of methanol production, achieving high selectivity, current density and stability simultaneously remains a grand challenge[8]. To date, there have been many efforts to achieve high methanol selectivity by constructing efficient electrocatalysts, such as metal alloys[9], metal chalcogenides[10], single-atom materials[11], metal-organic compounds[12], molecular catalysts[13], and the pyridine-based catalysts[14]. Methanol could be produced on isolated Cu decorated carbon nanofibers with Faradaic efficiency (FE) of 44% and a current density of 93 mA cm$^{-2}$ [11]. Boron phosphide exhibited methanol selectivity up to 92%, but the current density was only 0.2 mA cm$^{-2}$ [15]. Pd-Cu aerogel has also been employed as an electrocatalyst for electrochemical synthesis of methanol with FE of 80% and a current density of 31.8 mA cm$^{-2}$ [9]. Besides, Cu selenide catalyst could boost CO$_2$ reduction to methanol with a FE of 77.6% and a current density of 41.5 mA cm$^{-2}$ [16]. Generally, methanol FE reported is lower than 50 % when the current density is higher than 50 mA cm$^{-2}$ (please see the details in Supplementary Table 1)[5,9,13,16–18]. There is no doubt that the design of a robust electrocatalyst for CO$_2$-to-methanol is highly desired.

To date, the most efficient catalysts reported for producing methanol are Cu-based materials[19]. However, the insufficient stability of Cu-based materials under reductive conditions led to continuously diminished methanol yield. Doping may create high-energy surfaces for catalysis and improve the efficiency of methanol generation[20,21]. Some doping methods with different doping components have been reported to improve the CO$_2$-to-methanol activity, such as organically doped Cu-Pt alloy[22], Cu doped Pd aerogels[9], and O decorated Cu electrode[23]. Furthermore, mechanistic studies show that doping can modulate the surface electronic structure of the catalysts and their interaction with the reaction intermediates through lattice strain and coordination effect, leading to the improved activity[24–26]. However, the principle of competition among different components (active site, host, and doping element) remains elusive[9,16], and the efficiency to generate methanol is still relatively low[5,27]. Consequently, the main obstacle is how to achieve controllable coupling between different components to regulate the catalytic activity and selectivity.

Dual doping is of considerable interest, which can exploit the synergistic effect of the beneficial influences of the different heteroatoms[28,29]. The dual doping provides a basis for creating more lattice defects, vacancies, and active sites to control catalytic activity[30,31]. Typically, anion and cation possess opposite charge states, and their dual doping can bring more space to regulate the activity of the catalyst, which has a wide application in electrochemical hydrogen evolution reaction (HER), oxygen evolution reaction, and oxygen reduction reaction[32–34]. In the light of these examples in electrochemical reactions, we propose that dual doping also has the potential for designing efficient electrocatalysts for CO$_2$RR[35]. Against this backdrop, we sought to manipulate cation and anion doping pairs in electrocatalysts to simultaneously address the following challenges, (i) suppressing HER to improve CO$_2$RR activity, (ii) manipulating the kinetics of the intermediates to enhance methanol selectivity, and (iii) changing the intrinsic properties of catalysts to increase durability.

Here we present an in situ dual-doping strategy to construct a class of efficient CO$_2$-to-methanol electrocatalysts. In this approach, we use cation (Ag, Au, Zn, Cd) and anion (S, Se, I) doping to study the influence of dual doping in Cu$_2$O/Cu host on the CO$_2$-to-methanol reaction performance. Taking Ag and S dual doping as example, the Cu atom near the heteroatoms (Ag and S) in the interface structure of the host (Cu$_2$O/Cu) can effectively serve as the active center for methanol production. The density functional theory (DFT) calculations demonstrate that the anion S regulates the electronic structure of the adjacent Cu atom facilitating the formation of *CHO from *CO and the cation Ag mainly increases the reaction barrier of HER. Their synergistic interactions with the host material enhance the CO$_2$RR to methanol. Remarkably, Ag and S co-doped Cu$_2$O/Cu (Ag,S-Cu$_2$O/Cu) achieve a maximum methanol FE of 67.4% at the potential of −1.18 V vs. reversible hydrogen electrode (RHE) with a high current density of 122.7 mA cm$^{-2}$ in an ionic liquid (IL)/H$_2$O electrolyte.

## Results

**Synthesis and morphology of dual-doping catalysts**. To improve the kinetics of CO$_2$RR-to-methanol reaction, we design a series of dual-doping catalysts. Cu$_2$O/Cu was served as the host structure (Cu$_2$O/Cu host), and it was doped with various cations (x = Ag, Au, Zn, Cd) and anion (y = S, Se, I), which yielded the dual-doping structures (denoted as x,y-Cu$_2$O/Cu). Figure 1a and Supplementary Fig. 1 show the representative in situ dual-doping synthesis process of Ag, S-Cu$_2$O/Cu. In this process, the Cu$_2$S thin film was first synthesized on the Cu foam substrate (Fig. 1b) by the electrochemically assisted assembly technique. The anode (Cu foam) acted as the Cu$^+$ source, and the S$^{2-}$ ion in the electrolyte could bond to Cu$^+$ near the anode when a bias potential was applied. Surfactant hexadecyl trimethyl ammonium bromide (CTAB) was intentionally added to the electrolyte, which acted as a structure-directing agent for regulating the structure of the Cu$_2$S (Supplementary Fig. 2, scanning electron microscopy (SEM)). Subsequently, a certain amount of Ag$^+$ solution was introduced on the Cu$_2$S electrode to obtain Ag-Cu$_2$S precursor without morphology change (Fig. 1c and Supplementary Fig. 2f). The Ag-Cu$_2$S was then in situ transformed into Ag,S-Cu$_2$O/Cu in a CO$_2$-saturated 1-butyl-3-methylimidazolium tetrafluoroborate (BMImBF$_4$, an IL)/H$_2$O electrolyte. SEM images reveal that the in situ generated Ag,S-Cu$_2$O/Cu had a typical three-dimensional (3D) porous architecture with an interconnected network of nanowires (Fig. 1d, e). To demonstrate that this synthesis method is universal, other doping Cu$_2$O/Cu materials from mono to dual doping were also prepared (Supplementary Figs. 3, 4, and 5). The SEM results suggest that anion S played an important role in the formation of the porous nanonetwork architecture.

We then performed a detailed observation of the in situ transformation process (Supplementary Fig. 6). The SEM and transmission electron microscopy (TEM) images of the catalysts obtained at different electrochemical reduction times are presented in Supplementary Figs. 7 and 8, demonstrating that Ag-Cu$_2$S precursor can in situ transform into 3D porous Ag,S-Cu$_2$O/Cu nanonetwork, via Ag-Cu$_2$S/Cu$_2$O intermediate. Results indicate that the Ag,S-Cu$_2$O/Cu porous nanonetwork structures were completely formed after a reduction time of 10 min. High-resolution transmission electron microscopy (HRTEM) images show that the lattice spacings of Ag,S-Cu$_2$O/Cu were 0.18 nm and 0.25 nm, corresponding to the lattice plane distance of (200) plane of face-centered cubic Cu and (111) plane of cubic Cu$_2$O,

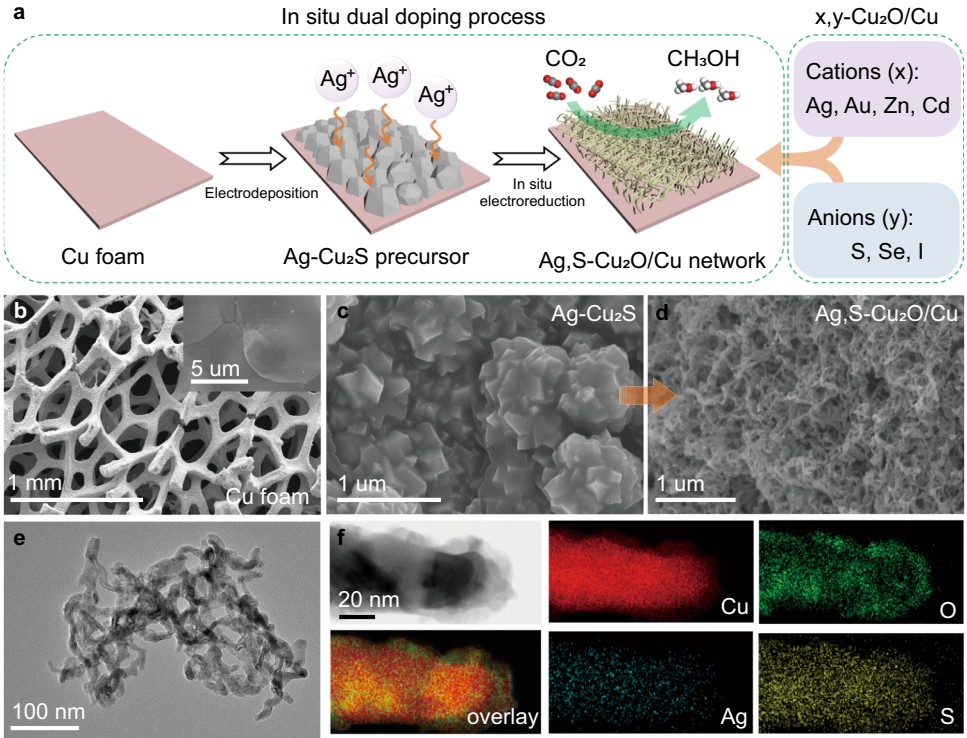

**Fig. 1 Synthesis and structure characterizations of x,y-Cu₂O/Cu. a** Schematic diagram of the in situ dual-doping process for preparing the x,y-Cu₂O/Cu catalysts. SEM images of **b** bare Cu foam substrate, **c** Ag-Cu₂S precursor on Cu foam substrate, and **d** Ag,S-Cu₂O/Cu formed at electroreduction time of 30 min. **e** TEM image of the Ag,S-Cu₂O/Cu. **f** Scanning transmission electron microscopy (STEM) and elemental mapping of a typical Ag,S-Cu₂O/Cu.

respectively (Supplementary Fig. 8f). The energy-dispersive X-ray spectroscopy (EDS) mapping analysis (Fig. 1f) further confirmed the uniform dispersion of Cu, Ag, S, and O species in the Ag,S-Cu₂O/Cu. The time-dependent X-ray diffraction (XRD) and Raman spectra also show that the catalyst after in situ conversion was Ag, S dual-doped Cu₂O/Cu (Supplementary Figs. 9 and 10). The Ag,S-Cu₂O/Cu spectra become unchanged after 10 min of conversion, which is consistent with the analysis of SEM and TEM. In detail, without Ag doping, the XRD patterns of Cu₂S precursor consisted of two crystalline phases, indexing to Chalcocite (PDF#26-1116) and Chalcocite-M (PDF#33-0490) (Supplementary Fig. 11a). After introducing Ag⁺ into Cu₂S, the intensities of Chalcocite-M diffraction peaks were improved, indicating that the interaction between Ag⁺ and Cu₂S facilitating the anchoring of Ag element. In addition, Cu₂O (PDF#05-0667) and Cu (PDF#04-0836) diffraction peaks can be found after in situ reductions and no Cu₂S peaks are observed (Supplementary Fig. 11b). In the Raman spectrum of Cu₂S (Supplementary Fig. 10), the intensity of 475 cm⁻¹ peaks became weak after Ag doping. After the in situ conversion, Ag ions were doped into the Cu₂O causing its lower crystallinity (Supplementary Fig. 11b).

**Electronic structure.** The X-ray photoelectron spectroscopy (XPS) analysis was performed to further investigate the surface chemical composition and elemental valence states of the catalysts. The peaks at 932.1 eV (Cu $2p_{3/2}$) and 952.0 eV (Cu $2p_{1/2}$) retained the characteristic feature of Cu species (Supplementary Fig. 12)[36]. This is further confirmed by Auger electron spectroscopy (AES) that the Cu species of Ag,S-Cu₂O/Cu were mainly composed of Cu (0) and Cu (I), and Cu (I) was predominant (Supplementary Fig. 13a)[37]. In comparison with Ag-Cu₂S, the binding energies of Cu⁺ LMM Auger peak and Ag⁺ $3d_{5/2}$ (Supplementary Fig. 13b) in Ag,S-Cu₂O/Cu were positively shifted by 0.5 eV and 0.3 eV, respectively. It indicates that the Cu and

Ag sites were in electron-deficient states, which means that the Cu and Ag mainly possessed oxygen coordination in Ag,S-Cu₂O/Cu, whereas they exhibit the absolute sulfur coordination environment in the Ag-Cu₂S precursor[38]. This is in good agreement with the XRD analysis. The peaks at around 162 eV (S $2p$) are belonging to the S²⁻ (Supplementary Fig. 13c)[39]. The dramatic decrease in the intensity revealed that the amount of S²⁻ was significantly reduced after in situ conversion. From the peak of O 1 s spectrum (Supplementary Fig. 13d), it is obvious that the lattice oxygen of Cu₂O existed in the catalyst[40]. The quasi-in situ XPS revealed the existence of Ag and S species during the electrochemical reduction process, and the atomic ratios of Ag/Cu and S/Cu were unchanged after 10 min in situ conversion (Supplementary Fig. 14). Moreover, the inductively coupled plasma (ICP) analysis verified the catalyst components. As shown in Supplementary Table 2, the atomic contents of Cu, Ag, and S in Ag,S-Cu₂O/Cu were 74.9%, 2.3%, and 5.2%, respectively. Further estimations show that the molar ratio of metallic Cu and Cu₂O is about 1.1. The result is in consistent with the quasi-in situ XPS analysis.

**Electrocatalytic CO₂RR performance.** The Ag,S-Cu₂O/Cu catalyst was firstly tested for CO₂RR in 1-butyl-3-methylimidazolium tetrafluoroborate (BMImBF₄)/H₂O (molar ratio 1:3) electrolyte using a typical H-type cell. In this study, the linear sweep voltammetry (LSV) curves over various electrodes were determined (Fig. 2a), including non-doped Cu₂O/Cu, mono-doped S (or Ag)-Cu₂O/Cu, and dual-doped Ag,S-Cu₂O/Cu. Over Ag,S-Cu₂O/Cu, current density was much higher in CO₂-saturated electrolyte than that in N₂-saturated electrolyte in the potential range from −0.58 V to −1.38 V vs. RHE, indicating the occurrence of CO₂RR. Noting that the Ag,S-Cu₂O/Cu electrode exhibited a more positive potential of −1.09 V vs. RHE than other electrodes at the current density of 100 mA cm⁻², we assume that Ag and S

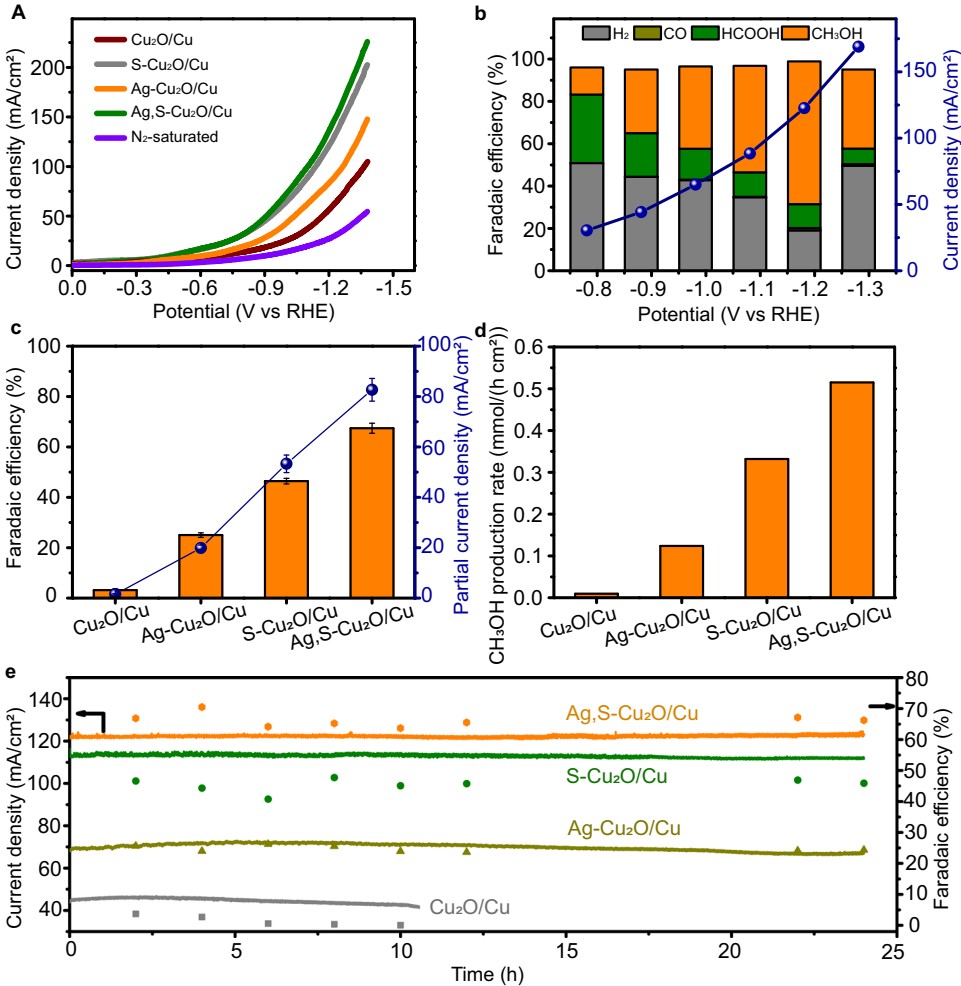

**Fig. 2 High $CO_2$-to-methanol performance of Ag,S-$Cu_2O$/Cu electrocatalyst. a** Linear sweep voltammetry (LSV) curves of various catalysts in $CO_2$-saturated or $N_2$-saturated $BMImBF_4/H_2O$ (mole ratio is 1:3) electrolyte with the scan rate of 10 mV s$^{-1}$. **b** Potential-dependent product selectivity and total current density for $CO_2RR$ by Ag,S-$Cu_2O$/Cu electrode. **c** The FE of methanol and corresponding partial current density of $CO_2RR$ catalyzed by different catalysts. Error bars represent the standard deviations from multiple measurements. **d** Methanol production rates over different catalysts. **e** Long-term stability over different catalysts. Electrolysis experiments were carried out at −1.18 V vs. RHE.

dual doping may increase the number of active sites that were kinetically favorable for $CO_2RR$. To verify the hypothesis, we conducted the electric double-layer capacitance ($C_{dl}$), which was calculated to estimate the electrochemical active surface area (ECSA) of various catalysts (Supplementary Fig. 15)[3]. The linear slopes in Supplementary Fig. 16 show that the Ag,S-$Cu_2O$/Cu and S-$Cu_2O$/Cu had a larger ECSA, indicating that S doping is responsible for the morphology architecture and Ag doping might further steer kinetics of $CO_2RR$ via generating more active sites. After normalizing the current density to ECSA (Supplementary Fig. 17), Ag,S-$Cu_2O$/Cu still exhibited the largest current density (80.2 mA cm$^{-2}$) at the potential of −1.18 V vs. RHE, which indicates that the dual doping could also improve the intrinsic activity of the catalyst.

The electrolysis performances at different applied potentials are displayed in Fig. 2b. It can be found that the Ag,S-$Cu_2O$/Cu mainly yielded $H_2$, CO, HCOOH, and methanol with a combined FE of around 100%. As the potential became more negative, the FEs of $H_2$ and HCOOH were gradually suppressed and that of methanol was increased. At the potential of −1.18 V vs. RHE, the current density over Ag,S-$Cu_2O$/Cu could reach 122.7 mA cm$^{-2}$ with a maximum methanol FE of 67.4%. For non-doping $Cu_2O$/Cu catalyst, the FE of methanol was only 3.5% with a limited

partial current density of 1.5 mA cm$^{-2}$ (Fig. 2c). A partial current density ($j_{methanol}$) of 82.7 mA cm$^{-2}$ was achieved over the Ag,S-$Cu_2O$/Cu, which is roughly 55, 4, and 1.5 times larger than that of $Cu_2O$/Cu, Ag-$Cu_2O$/Cu, and S-$Cu_2O$/Cu, respectively. At the optimized condition, the methanol production rate over Ag,S-$Cu_2O$/Cu electrode could reach 0.52 mmol h$^{-1}$ cm$^{-2}$ (Fig. 2d). Systematic comparisons to state-of-the-art catalysts reveal that this method can construct very efficient $CO_2$-to-methanol electrocatalysts, while the FE was generally below 50% when the current density was higher than 50 mA cm$^{-2}$ over the reported catalysts (Supplementary Table 1). Long-term electrolysis was also performed to verify the stability of the catalysts. As shown in Fig. 2e, no obvious decays were observed in both methanol FE and current density over Ag,S-$Cu_2O$/Cu, S-$Cu_2O$/Cu, and Ag-$Cu_2O$/Cu, while the $Cu_2O$/Cu catalyst almost lost its catalytic ability toward methanol, approaching 0 % after 10 h. After the continuous $CO_2$ electrolysis, the elemental valence states (Supplementary Fig. 18) and the morphology structures (Supplementary Fig. 19) of the Ag,S-$Cu_2O$/Cu were well preserved. This indicates that dual doping could also enhance the stability of the catalysts. The in situ strategy could form highly dispersed and adhesive doping catalysts on the wall of the Cu foam, maintaining a high current density for long-term electrolysis.

In order to verify that the product was derived from $CO_2RR$, we used isotope-labeled $^{13}CO_2$ or $N_2$ to replace $CO_2$ in the same set-up. The $^{13}C$ NMR spectra (Supplementary Fig. 20a–c) show two obvious peaks at 163.1 and 49.5 parts per million, which are attributed to $H^{13}COO^-$ and $^{13}CH_3OH$, respectively. From $^1H$ NMR spectra in Supplementary Fig. 20d,e, we can see the H signals of formate and methanol, which both split into two peaks by coupling with $H–^{13}C$ atom. These data confirm that the feeding $CO_2$ gas was the only source of carbon in the reduction products.

It is worth mentioning that electrolyte often plays an important role in $CO_2RR$. IL can greatly improve the solubility of $CO_2$ in the electrolyte and ensure the effective supply of $CO_2$ during the $CO_2RR$ process. However, the high viscosity of IL would lead to lower mass transport and thus decrease the $CO_2RR$ activity[41,42]. Therefore, we used $BMImBF_4/H_2O$ with different molar ratios to study the impact of electrolyte composition on product selectivity. From Supplementary Fig. 21, we can find that $BMImBF_4/H_2O$ with a molar ratio of 1:3 exhibited the highest current density and FE for methanol production. To further understand the role of electrolytes, we used ILs with different anions and cations for comparison. It can be observed that the imidazolium cations and fluorine-containing anions both influenced the methanol selectivity significantly (Supplementary Fig. 22). For cations, IL with two substituents in the imidazolium cation, such as $BMImBF_4$ and $EMImBF_4$, was beneficial to the reaction. This is because the spatial structure of the cation is conductive to the adsorption of $CO_2$ and the intermediates[41,43]. For anions, fluorine-containing anions, such as tetrafluoroborate ($BF_4^-$) and trifluoromethane-sulfonate ($OTF^-$), are more favorable for $CO_2RR$, which resulted partially from their suitable interaction with $CO_2$[44].

Based on these observations, we can conclude that the excellent performance of $Ag,S-Cu_2O/Cu$ electrode resulted partially from the structure construction via the in situ strategy. We find that the morphology and structure of the $Ag,S-Cu_2O/Cu$ could be tuned via the structural evolution of $Cu_2S$ with different amounts of CTAB (Supplementary Figs. 2 and 23). In the absence of CTAB, the $Ag,S-Cu_2O/Cu$ obtained from the $Cu_2S$ was a nanoparticle structure, which possessed only 30.5% FE of methanol. With the increasing amount of CTAB, porous $Ag,S-Cu_2O/Cu$ generated from $Cu_2S$ could reach 67.4% FE of methanol with a CTAB amount of 0.35 g (Supplementary Fig. 24). According to Supplementary Fig. 25, the $C_{dl}$ values of $Ag,S-Cu_2O/Cu$ with nanonetwork structure (0.35 g CTAB) was 37.2 mF cm$^{-2}$, which was larger than that of $Ag,S-Cu_2O/Cu$ with nanoparticle structure (0 g CTAB, 30.1 mF cm$^{-2}$). Therefore, we can know that the $Ag,S-Cu_2O/Cu$ with nanonetwork structure (0.35 g CTAB) had a larger ECSA. The porous nanonetworks resulted in more active

sites than nanoparticles for catalyzing the $CO_2RR$ to methanol. In addition, the electrochemical impedance spectrum (EIS, Supplementary Fig. 26) was carried out to probe the effect of structural features on the charge transport kinetics at the potential of $-1.18$ V vs. RHE. It showed that charges resistance ($R_{ct}$) on $Ag,S-Cu_2O/Cu$ was much lower than that on other catalysts, which indicates favorable kinetics on $Ag,S-Cu_2O/Cu$ towards $CO_2RR$. Therefore, the hierarchical structure was crucial to $CO_2$ activation and intermediate stabilization, resulting in higher activity and selectivity.

The current densities and methanol FEs are also strongly dependent on the amount of $Ag^+$ (Supplementary Fig. 27). As the feed amount of $Ag^+$ increased from 2 to 6 µmol, the current density gradually increased. Methanol FE reached a maximum of 67.4% at 4 µmol, after which it dropped with increasing FE of $H_2$ and CO. The main reason for this is that the excess amount of $Ag^+$ ions agglomerated into Ag metal particles (Supplementary Fig. 28, XRD), promoting the HER and CO pathway.

We turn now to screen the effective doping pairs, the $CO_2RR$ performance tests of $Ag,Se-Cu_2O/Cu$, $Ag,I-Cu_2O/Cu$, $Au,S-Cu_2O/Cu$, $Cd,S-Cu_2O/Cu$, and $Zn,S-Cu_2O/Cu$ were conducted, and the results are showed in Fig. 3a. It indicates that Se or I doping exhibited a poor methanol selectivity relative to S doping. HER could be prohibited by switching of anodic dopants, following a decreasing sequence of $Ag > Au > Cd > Zn$. Therefore, we consider that the kinetic of the $CO_2RR$ to methanol pathway was sensitive to the doping pairs.

**DFT calculations.** To further confirm our assumption, we explored the relationship between heteroatoms and catalytic activity from a theoretical viewpoint (Supplementary Table 3). Based on our previous results, we build the $Cu_2O/Cu$ host structure by loading Cu cluster on the $Cu_2O$ (111) surface, which is shown in Supplementary Fig. 29a. The Cu and O atoms on $Cu_2O/Cu$ were then replaced by cation and anion heteroatoms, respectively, to obtain the dual-doping structures (Supplementary Fig. 29b–d). We found that hydrogenation of *CO intermediate to form *CHO was an endothermic step on all the doping structures. Therefore, we plotted the calculated Gibbs free-energy difference values of $\Delta G_{*CHO}$ and $\Delta G_{*CO}$ along with the measured partial current density of methanol over the catalysts with different doping pairs (Fig. 3b). Interestingly, the plot shows a volcanic curve relationship between doping pairs and catalytic performance, and shows the general trend of the above $CO_2$ electrolysis results. When the difference value was close to about 0.65 eV, the partial current density of methanol reached the maximum over $Ag,S-Cu_2O/Cu$, which represented the high

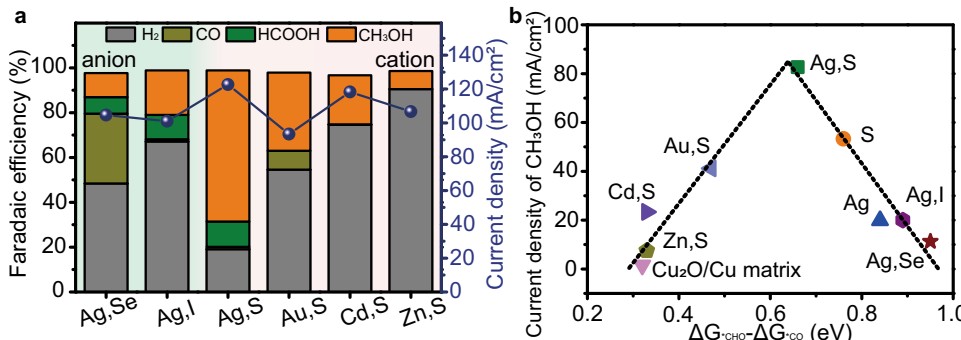

**Fig. 3 Anion-cation double doping effect. a** The performance of $CO_2RR$ catalyzed by different dual-doping catalysts at the potential of $-1.18$ V vs. RHE. **b** Plot of the experimentally measured partial current density of methanol versus the theoretically calculated Gibbs free-energy differences of $\Delta G_{*CHO}$ and $\Delta G_{*CO}$ for different doping catalysts.

activity of $CO_2$-to-methanol. For other mono or dual-doping catalysts, they appeared on both sides of the volcanic curve, and thus had lower activity (Changing the anion in doping pairs resulted in lower energy barriers (S < I < Se), and changing the cation could improve the partial current density of methanol (Ag > Au > Cd > Zn)). A reasonable explanation of the result is that the doping pairs could change the electronic structure and morphology, which are related to the methanol selectivity. Anion S could effectively adjust the adsorption space position of the *CHO intermediate with a lower formation energy barrier. Cation Ag could inhibit HER to further improve the selectivity and current density for methanol production.

In addition, we pursued theoretical insights into the relationship between heteroatoms and intrinsic $CO_2RR$ activity to deeply understand the effect of doping ions. Even though many bimetallic materials can act as tandem catalysts for the enhancement of $CO_2RR$ product selectivity, the Ag and Cu in our catalyst do not constitute a tandem catalyst, as confirmed by our experimental findings (Supplementary Fig. 30). Therefore, we consider that, at the dual-doping interface, the electronic interactions between Cu active center and *COOH intermediate change greatly, affecting the subsequent reaction path. The optimized adsorption configurations of reaction intermediate on the simulated interface structures are displayed in Supplementary Figs. 31–34. The transformation of surface charges after *COOH adsorption over these interface structures was theoretically investigated. As depicted in Supplementary Fig. 35, we can find that Cu active sites and O atoms attracted some electrons from C atoms in the optimized *COOH intermediate state. The formation of *COOH could reach a stable configuration with moderate free energy (0.47, 0.08, and 0.02 eV for Ag-$Cu_2O$/Cu, S-$Cu_2O$/Cu, and Ag,S-$Cu_2O$/Cu, respectively). This enables $CO_2$ transformation to more reduced products with a multi-electron process. However, strong adsorption of *COOH intermediate onto the $Cu_2O$/Cu interface ($\Delta G_{*COOH} = -2.87$ eV) led to the severe aggregation of *COOH intermediates at the active sites and blocked

the subsequent reaction path. Therefore, the dissociation of *COOH to form *CO was more likely to occur on the doped structures (Ag-$Cu_2O$/Cu, S-$Cu_2O$/Cu, and Ag,S-$Cu_2O$/Cu) than that on $Cu_2O$/Cu structure. Subsequently, the hydrogenation of *CO to *CHO is an endothermic process with the highest energy barrier in the methanol production process, representing the activity of the catalyst for methanol production. The heteroatoms doping could effectively adjust the adsorption space position of the *CHO intermediate and its O atom (Fig. 4a). Over Ag,S-$Cu_2O$/Cu, the barrier energy for the hydrogenation of *CO to *CHO (0.66 eV) is lower than that over Ag-$Cu_2O$/Cu (0.88 eV) and S-$Cu_2O$/Cu (0.76 eV) (Fig. 4b), indicating that the *CHO is easier to form on the surface of the dual-doping catalyst to further accept protons and electrons to form *$OCH_3$, then ended up with methanol.

Towards understanding the electronic effect on the binding strength of intermediates and the electronic structure of the catalysts, we carried out the density of states and Bader charge analysis. The projected density of state of the Cu in $Cu_2O$/Cu, Ag-$Cu_2O$/Cu, S-$Cu_2O$/Cu, and Ag,S-$Cu_2O$/Cu is shown in Supplementary Fig. 36. The Ag-$Cu_2O$/Cu ($-2.16$ eV, relative to the Fermi level), S-$Cu_2O$/Cu ($-2.15$ eV), and Ag,S-$Cu_2O$/Cu ($-2.14$ eV) catalysts have a lower d-band center than that of the $Cu_2O$/Cu ($-1.88$ eV) due to the doping effect and the local structural deformation[45,46]. It is reasonable to assume that a strong *COOH adsorption may lead to a trap of the intermediate on $Cu_2O$/Cu surface. Among these three doping catalysts, the Cu active centers near the heteroatoms in Ag,S-$Cu_2O$/Cu possess more electrons near the Fermi level than mono-doped catalysts (Ag-$Cu_2O$/Cu and S-$Cu_2O$/Cu). These differences reveal that the Ag,S-$Cu_2O$/Cu was the most moderate one in bonding *COOH, favoring the subsequent steps to form methanol (Supplementary Fig. 37). The Bader charge changes of each atom before and after doping were summarized in Supplementary Table 4, and the charge state changes around the heteroatoms were obvious. We also fitted a linear relationship between the oxidation state of the Cu atom

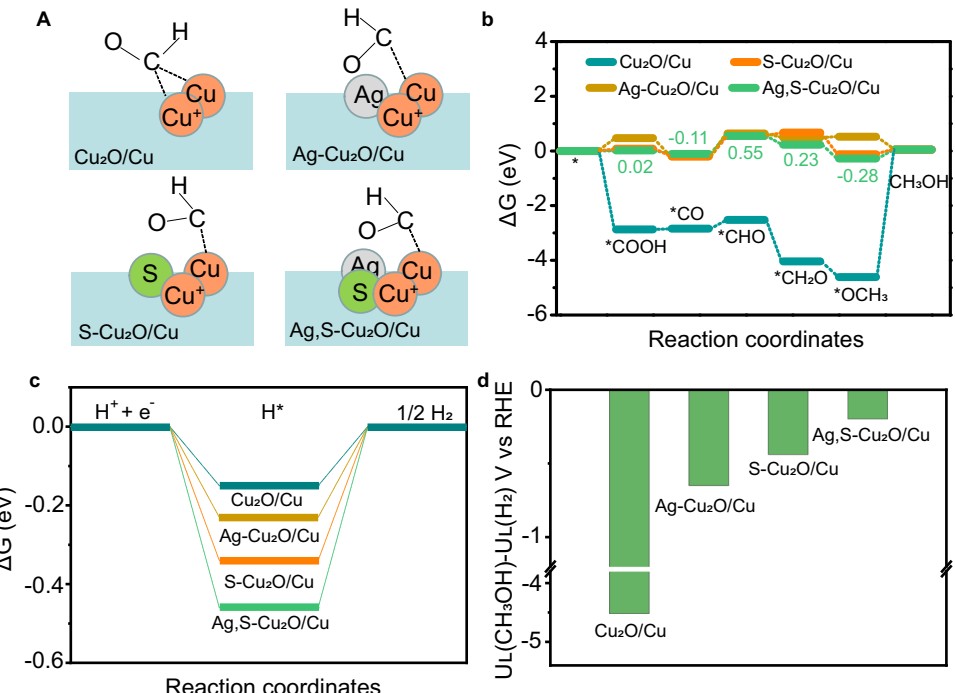

**Fig. 4 Theoretical investigation on the $CO_2$ reduction over Ag,S-$Cu_2O$/Cu. a** Schematic diagram of *CHO intermediate adsorption over $Cu_2O$/Cu, Ag-$Cu_2O$/Cu, S-$Cu_2O$/Cu, and Ag,S-$Cu_2O$/Cu. Reaction free-energy diagrams for the proposed steps of **b** $CO_2RR$ to methanol and **c** HER. **d** Differences in limiting potentials for $CO_2RR$ to methanol and HER over $Cu_2O$/Cu, Ag-$Cu_2O$/Cu, S-$Cu_2O$/Cu, and Ag,S-$Cu_2O$/Cu.

adjacent to the dopant and the Bader charge in Supplementary Fig. 38. The changes in the oxidation state of Cu indicate the interaction between Cu and heteroatom (Ag or S), as well as the surface geometrical changes (Fig. 1d, Supplementary Figs. 3 and 4). It should be noted that the oxidation states of Cu in Ag,S-$Cu_2O$/Cu were very similar to that of S-$Cu_2O$/Cu, confirming that the regulation of cation Ag on the electronic structure of Cu active center was inferior to that of anion S. Furthermore, the cationic and anodic effect can be mainly explained by the study of HER activity (Fig. 4c) and the limiting potentials difference between methanol path and HER over different electrodes (Fig. 4d). It is obvious that mono doping can increase the reaction barrier of HER ($-0.23$ eV and $-0.34$ eV for doping Ag or S, respectively, Fig. 4c) compared with non-doped $Cu_2O$/Cu ($-0.15$ eV). In addition, the HER is further suppressed over the Ag and S dual-doped structure ($-0.46$ eV). Noting that the difference between the limiting potentials for $CO_2RR$ and HER ($\Delta U = U_L(CO_2)$-$U_L(H_2)$, where $U_L = -\Delta G/e$) has been widely applied to describe the selectivity of the $CO_2RR$, and a more positive $\Delta U$ value denotes a higher selectivity. It is shown that Ag,S-$Cu_2O$/Cu had the largest positive value, followed by S-$Cu_2O$/Cu, Ag-$Cu_2O$/Cu, and $Cu_2O$/Cu, which clearly reveals that Ag,S-$Cu_2O$/Cu had the best performance for $CO_2$ selective conversion to methanol (Fig. 4d). The above results, taken together, suggest that the synergistic interaction between doping ions and the host materials can efficiently inhibit the HER activity and enhance the methanol selectivity.

## Discussion

We find that the dual doping of $Cu_2O$/Cu by suitable components is a promising strategy to enhance the $CO_2RR$ to methanol. The FE to methanol can achieve 67.4% with a very high current density of 122.7 mA cm$^{-2}$ over Ag,S-$Cu_2O$/Cu in the BMImBF$_4$/$H_2O$ binary electrolyte. The outstanding electrocatalytic performance of the dual-doping catalysts can be ascribed to the synergistic effect among dual doing pairs and $Cu_2O$/Cu host for producing methanol, as well as the three-dimensional porous architecture. The anion regulates the electronic structure of the adjacent Cu atom facilitating the formation of *CHO from *CO, whereas the cation mainly suppresses the HER, thus enhancing the kinetic process of $CO_2RR$ to methanol. We believe that the efficient and stable catalyst has the promising potential for application in the electrocatalytic reduction of $CO_2$ to methanol, and the in situ dual-doping strategy can also be used to design some other efficient electrocatalysts.

## Methods

**Materials**. KOH (A. R. grade), Na$_2$S·9H$_2$O (A. R. grade), $H_2SO_4$ (A. R. grade), lactic acid (A. R. grade), AgNO$_3$ (A. R. grade), KI (A. R. grade), NaSeO$_4$ (A. R. grade), and Cu foam (2 mm in thickness, purity > 99.99%) were provided by Sinopharm Chemical Reagent Co., Ltd, China. Hexadecyl trimethyl ammonium bromide (CTAB), SnCl$_2$ (A. R. grade), Cd(NO$_3$)$_2$·4H$_2$O (A. R. grade), Zn(NO$_3$)$_2$·6H$_2$O (A. R. grade), and HAuCl$_4$ (99.995%) were purchased from Aldrich. $CO_2$ and N$_2$ (Beijing Beiwen Gas Chemical Industry Co., Ltd., research grade) had a purity of 99.9999% and used as received. $^{13}CO_2$ (99 atom% $^{13}C$) was purchased from Sigma–Aldrich. Nafion N-117 membrane (0.180 mm thick, ≥0.90 meg/g exchange capacity) was purchased from Alfa Aesar China Co., Ltd. 1-butyl-3-methylimidazolium tetrafluoroborate (BMImBF$_4$, purity >99%), 1-ethyl-3-methylimidazolium tetrafluoroborate (EMImBF$_4$, purity >99%), 1-butyl-2,3-dimethylimidazolium tetrafluoroborate (BMMImBF$_4$, purity >99%), 1-butylimidazolium tetrafluoroborate (BImBF$_4$, purity >99%), 1-butyl-3-methylimidazolium trifluoromethanesulfonate (BMImOTf, purity >99%) and 1-butyl-3-methylimidazolium trifluoroacetate (BMImCF$_3$COO, purity >99%) were obtained from the Centre of Green Chemistry and Catalysis, Lanzhou Institute of Chemical Physics, Chinese Academy of Sciences. Aqueous solutions were prepared with deionized water (Millipore 18.2 MΩ cm).

**Synthesis of $Cu_2S$ on Cu foam**. The $Cu_2S$ precursor on Cu foam was prepared via a simple electrochemically assisted assembly method. Before synthesis, the Cu foam surface was initially cleaned by acid wash (0.5 M $H_2SO_4$) in an ultrasound bath for

30 min. Then the Cu foam was washed with deionized water for five times before use. A typical cyclic voltammetry (CV) was conducted in an H-type glass cell, with two electrolyte zones being separated by a Nafion 117 membrane using saturated calomel electrode (SCE) as reference electrode and Pt gauze as a counter electrode. The electrolyte contained 3.2 M KOH, 2.3 M lactic acid, and 0.1 M Na$_2$S·9H$_2$O (without or with CTAB). The cell was placed in a water bath of 40 °C. For a typical synthesis of $Cu_2S$, the amount of CTAB in 40 mL electrolyte was 0.35 g. The CV was conducted from $-0.2$ V to $-0.8$ V vs. SCE with a scan rate of 5 mV/s for 10 cycles. After CV, the working electrode was washed with deionized water for three times for further use. The $Cu_2O$ precursor was synthesized by the same method without Na$_2$S·9H$_2$O.

**Preparation of Ag-$Cu_2S$ or Ag-$Cu_2O$**. When Ag was used, 34 mg of AgNO$_3$ was dissolved in 10 mL of deionized water. A certain amount of AgNO$_3$ solution was evenly deposited dropwise to the $Cu_2S$ or $Cu_2O$ precursor and dried under an infrared (IR) lamp to prepare Ag-$Cu_2S$ or Ag-$Cu_2O$.

**Preparation of Ag,S-$Cu_2O$/Cu**. Ag,S-$Cu_2O$/Cu was prepared by an electrochemical reduction method. The electrochemical reduction was conducted at ambient temperature in a typical H-type cell. The Ag-$Cu_2S$ was used as the working electrode. A BmimBF$_4$/$H_2O$ solution with the mole ratio of 1:3 was used as a cathode electrolyte with saturated $CO_2$. 0.5 M $H_2SO_4$ aqueous solution was used as an anodic electrolyte. For a typical synthesis of Ag,S-$Cu_2O$/Cu, the electrochemical reduction was performed by applying a potential of $-1.6$ V vs. SCE for 30 min with $CO_2$ bubbling under continuous stirring. The $Cu_2O$/Cu, S-$Cu_2O$/Cu, and Ag-$Cu_2O$/Cu were fabricated by the same procedure using $Cu_2O$, $Cu_2S$, and Ag-$Cu_2O$, respectively, as the working electrode.

**Preparation of other x,y-$Cu_2O$/Cu**. Other doping catalysts were prepared following the same procedure except for using corresponding chemicals. Using KI and NaSeO$_4$ to replace Na$_2$S·9H$_2$O could synthesize Ag,I-$Cu_2O$/Cu and Ag,Se-$Cu_2O$/Cu, respectively. Using SnCl$_2$, Cd(NO$_3$)$_2$·4H$_2$O, Zn(NO$_3$)$_2$·6H$_2$O, and HAuCl$_4$ to replace AgNO$_3$ could obtain dual-doping Sn,S-$Cu_2O$/Cu, Cd,S-$Cu_2O$/Cu, Zn,S-$Cu_2O$/Cu, and Au,S-$Cu_2O$/Cu, respectively.

**Material characterization**. The morphologies of materials were characterized by a HITACHI S-4800 scanning electron microscope (SEM) and a JEOL JEM-2100F high-resolution transmission electron microscopy (HRTEM). Powder X-ray diffraction (XRD) patterns were acquired with an X-ray diffractometer (Model D/MAX2500, Rigaka) with Cu-Kα radiation, and the scan speed was 5º/min. For XRD measurements, in order to get a sufficient sample, several electrodes were prepared under the same conditions and the catalysts were scraped and collected for characterization. X-ray photoelectron spectroscopy (XPS) analysis was conducted on the Thermo Scientific ESCALab 250Xi (USA) using 200 W monochromatic Al Kα radiation. The 500 μm X-ray spot was used for XPS analysis. The base pressure in the analysis chamber was about $3 \times 10^{-10}$ mbar. Raman spectroscopy (Horiba Labram HR Evolution Raman System) was conducted using a 785 nm excitation laser and signals were recorded using a 20 s integration by averaging two scans.

The quasi in situ X-ray photoelectron spectra (XPS) were measured on an AXIS ULTRA DLD spectrometer with Al K$_\alpha$ resource ($hv = 1486.6$ eV). The samples were prepared in a glove box filled with nitrogen and transferred to the XPS chamber for measurement. For investigating the evolution of Cu, Ag, and S species in the reaction process, catalysts were electrolyzed at different times in the $CO_2$-saturated electrolytes. After that, samples were washed with acetone immediately and put into the glove box. Then, the samples were cut into 5 × 5 mm and glued on a stage with a double-sided adhesive. The stage was evacuated under a vacuum to prevent the samples to be oxidized in the air.

**Electrocatalytic $CO_2$ reduction**. An electrochemical workstation (CHI 660E, Shanghai CH Instruments Co., China) was used for the electrochemical experiment. The LSV measurements and controlled potential electrolysis were carried out in a typical H-type cell. The as-synthesized electrode was used as the working electrode. The SCE was used as the reference electrode and Pt gauze was used as the counter electrode. The cathode and anode compartments were separated through a Nafion 117 proton exchange membrane. A BMImBF$_4$/$H_2O$ with a mole ratio of 1:3 was used as a cathode electrolyte. To study the effect of electrolytes, the BMImBF$_4$ was replaced by other ILs (EMImBF$_4$, BMMImBF$_4$, BImBF$_4$, BMImOTf, and BMImCF$_3$COO). 0.5 M $H_2SO_4$ aqueous solution was used as an anodic electrolyte. Under continuous stirring, $CO_2$ was bubbled into the catholyte for 30 min before electrolysis. After that, electrochemical $CO_2$ reduction was carried out with $CO_2$ bubbling (20 mL min$^{-1}$). The potentials were converted to the reversible hydrogen electrode (RHE) reference scale using the relation
$E_{RHE} = E_{SCE} + 0.244 + 0.059 \times pH$.

**Double-layer capacitance ($C_{dl}$) measurement**. The cyclic voltammetry measurement was conducted in BMImBF$_4$/$H_2O$ electrolyte with the mole ratio of 1:3 using a three-electrode system using the as-prepared electrode as a working cathode. Cyclic voltammogram measurements of the catalysts were conducted from $-0.5$ to $-0.6$ V vs. SCE with various scan rates to obtain the double-layer capacitance ($C_{dl}$) of

different catalysts. The $C_{dl}$ was estimated by plotting the $\Delta j$ ($j_a$-$j_c$) at −0.55 V vs. SCE against the scan rates, in which $j_a$ and $j_c$ are the anodic and cathodic current densities, respectively. The linear slope was equivalent to twice of the $C_{dl}$.

**Electrochemical impedance spectroscopy (EIS) measurement.** The EIS measurement was conducted in a $CO_2$-saturated $BMImBF_4$/$H_2O$ electrolyte with the mole ratio of 1:3 at the potential of −1.18 V vs. RHE with an amplitude of 5 mV of 0.01 to $10^4$ Hz.

**Product analysis.** The gaseous product of the electrochemical experiment was collected using a gasbag and analyzed by gas chromatography (GC, HP 4890D). The liquid product was analyzed by $^1$H NMR (Bruker Avance III 400 HD spectrometer) in Acetonitrile-d3 with TMS as an internal standard. The FEs of the products were calculated using the amounts of the products obtained from GC and $^1$H NMR analysis. The ILs was stable in this work. In $^1$H NMR spectra, the C(2)-H on the cation of ILs was used as the internal standard. Because the concentration of IL was known, the relative peak area of HCOOH/C(2)-H or $CH_3OH$/C(2)-H can be calculated.

**Theoretical calculations.** All Density functional theory calculations were carried out by Vienna Ab-Initio Simulation Package (VASP)[47]. The projector augmented plane wave (PAW) pseudopotential basis set and generalized gradient approximation (GGA) functional by the PBE[48] parametrization were employed in these calculations. The energy cutoff was set to 400 eV. A gamma Monkhorst-Pack k-point sampling was employed for slab optimization and gas adsorption. A vacuum of at least 15 Å was adopted along the z-axis. During structure optimization, all energy change criterion was set to $10^{-4}$ eV in the iterative solution of the Kohn-Sham equation, and the atoms were relaxed until the force acting on each atom was less than 0.03 eV Å$^{-1}$. The long-range van der Waals interaction is described by the DFT-D3 approach[49]. According to our experimental results, we demonstrated the doping effect over the $Cu_2O$/Cu interface structure on methanol product intermediates. We calculated several structures with a Cu cluster supported on the $Cu_2O$, after optimization we obtained the most stable interface structure (Supplementary Fig. 29). The adsorption structures are considered in these stable interface structures. Due to the presence of highly unsaturated coordination atoms at the interface, we chose them as the potential active sites. As the adsorption intermediates have a certain spatial structure, resulting in fewer sites that can satisfy the adsorption conditions. We compared different adsorption sites and finally selected the most stable adsorption intermediate structure with the lowest energy, as shown in Supplementary Figs. 31–34. The computational hydrogen electrode method proposed by Norskov's group[50] was used to calculate the free energy of $CO_2$RR. The free energy was obtained from G = E + ZPE -TS + H, where E is the total energy, H is the enthalpy, S is the entropy and ZPE is the zero-point energy at room temperature (T = 298 K). The detailed values are displayed in Supplementary Table 3. The G(T) in Supplementary Table 3 represents ZPE -TS + H(T).

The $CO_2$ reduction to methanol was proposed via the following elementary steps:

$$* + CO_2 + H^+ + e^- \rightarrow *COOH \tag{1}$$

$$*COOH + H^+ + e^- \rightarrow *CO + H_2O \tag{2}$$

$$*CO + H^+ + e^- \rightarrow *CHO \tag{3}$$

$$*CHO + H^+ + e^- \rightarrow *CH_2O \tag{4}$$

$$*CH_2O + H^+ + e^- \rightarrow *OCH_3 \tag{5}$$

$$*OCH_3 + H^+ + e^- \rightarrow * + CH_3OH \tag{6}$$

where * denotes the active site in the interface structure.

## Data availability

The data that support the plots within this paper and supplementary information of this study are available in the Source data file. Additional data available from authors upon request. Source data are provided with this paper.

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

## Acknowledgements

We thank the National Key Research and Development Program of China (2017YFA0403003, 2020YFA0710203, 2017YFA0403101, and 2017YFA0403102), National Natural Science Foundation of China (22102192, 22022307, 22033009, 21890761,21733011, and 22121002), Chinese Academy of Sciences (QYZDY-SSW-SLH013), and the China Postdoctoral Science Foundation (BX20200336 and 2020M680680).

## Author contributions

P.S.L., Q.G.Z., and B.X.H. proposed the project, designed the experiments, and wrote the manuscript. P.S.L. performed the whole experiments. J.H.B., J.Y.L., C.J.C., X.F.S., and J.L.Z. performed the analysis of experimental data. Q.G.Z. and B.X.H. co-supervised the whole project. All authors discussed the results and commented on the manuscript.

## Competing interests

The authors declare no competing interests.
