## [Peer review file · Nature Communications]

REVIEWER COMMENTS

Reviewer #1 (Remarks to the Author):

In the present manuscript entitled “In situ dual doping for constructing efficient CO₂-to-methanol electrocatalysts”, the authors describe an investigation into a new oxide-derived copper structure utilizing dopant atoms to improve the CO₂RR activity and selectivity towards methanol. The authors find experimentally that Ag/S doped OD copper shows a higher selectivity towards methanol compared to other, previous approaches. They try to rationalize this with DFT calculations on somewhat related surface structures.

In the following, I will mostly focus on the DFT modelling of the system, since this coincides with my experience. The experimental results appear to be reasonably well executed and described; however, I would leave that evaluation to an experimental electrochemist. Lastly, I found the choice of the “SCE” as a (reported) reference electrode, instead of the standard hydrogen electrode rather counterproductive, since it makes the comparison to other results and the computational treatment actively more complicated.

Most of the manuscript is well written and argued, however, the computational part has several severe issues that the authors would have to address before any publication could even be considered. This includes the fact that the reported Gibbs adsorption energies in the SI do not reflect the displayed values in Figure 4b) which makes the conclusions of the paper questionable, at best. Lastly, large parts of the paper become unintelligible without constantly looking at SI figures, since the authors often do not even summarize the findings of the figures in the SI, which is against the idea of an SI.

I would either suggest to reject and resubmit or to perform major revisions to address the points below.

In the following I will detail larger comments and smaller remarks (mostly spelling, grammar etc):

Comments:

- 1) In the introduction the authors state “and its production results in large amount of carbon emission globally.” – Could the authors please provide a citation for this, ideally including an estimation of the magnitude of these emissions?
- 2) The authors state “Methanol Faradaic efficiency (FE) reported is lower than 50 % as the current density is higher than 50 mA/cm² (Table S1).” – My initial request would have been to put the current number into context with other publications, however, table S1 already does that. It would be good to state this and to cite the publications that were used to generate table S1.
- 3) It would be good to explain what the authors mean with the word “matrix”, since I would have expected the expression “host” in its stead. I am not against them using the word, however, please define it when it first appears.

- 4) Could the authors please provide their potentials vs standard hydrogen electrode, as is the de-facto standard in electrochemistry, rather than the obscure “saturated calomel electrode”.
- 5) There is little to no mention of DFT and its state of the art in the introduction.
- 6) The Figures and Tables in the SI should be ordered in the order they appear in the main text. Alternatively, please provide a table of contents with links to each table and figure for easy access.
- 7) The resolution of all figures is very low in the version that I reviewed. This can happen during the PDF generation process for the reviewers, but I would ask the authors to please check if they can provide the subfigures at a higher resolution or, ideally, provide them as vector graphics.
- 8) Could the authors please comment on why they only tested the reference Cu₂O/Cu system for around 10 hours and all the others for up to 25 hours (Figure 2e)?
- 9) Could the authors please explain what the units and calculation method is for the data shown in Figure S14? It is certainly not Ag/Cu = 0.5.

Also, the amount of Ag quickly drops by 75% and the amount of S drops by 90% ... I am not sure I would call that immediately "stable", however it seems to stabilize long-term.

- 10) The authors state “more active sites than nanoparticles for catalyzing the CO₂RR to methanol.” – Could the authors please provide data to back this up? E.g. EC surface area?
- 11) Given that the authors generate an Ag/Cu system: Could the authors please comment on how they make sure that they do not have an Ag/Cu tandem catalyst on the atomic level?
- 12) While I know that oxygen atoms can be present in OD copper under normal CO₂RR conditions (~1-1.5V vs RHE), could the authors please find some justification why this is should still be the case at almost 2V vs RHE or more.
- 13) The resolution of the adsorbate structures in the SI figures is significantly too low to understand the adsorption structure.
- 14) The adsorption structures that were used in the DFT simulations need to be provided as part of the SI.
- 15) The authors make no note of how they chose these, very specific adsorption structures. Given that the surfaces they chose are highly undercoordinated a significant justification for their choice, over e.g. doped Cu(111) would be needed.
- 16) Could the authors please provide a table with the charges assigned to each atom to showcase more clearly how the local charge distribution changes?
- 17) The authors claim that the “strong binding energy of *COOH onto the Cu₂O/Cu interface (-2.87 eV) leaded[SIC] to the severe aggregation of *COOH intermediates at the active sites and blocked the subsequent reaction path.”

This is not in agreement with previous studies and the adsorption energy also does not coincide with the number the authors report in the SI, table S2 (G(T),COOH,Cu₂O/Cu = 0.504eV), especially when viewed against the reported values for other surfaces (G(T), COOH, Ag/S/Cu₂O/Cu = 0.502eV)

18) The authors claim "Taking into account that the Cu₂O/Cu has a much higher d-band centre (relative to the Fermi level, -1.88 eV) compared with the doping catalysts (Figure S31)," – Firstly, this is, again, not readable without the SI, since the authors do not provide the comparison figure and leave the reader to go to the SI to look it up. Also, the introduction of 1-2 dopant atoms into a structure should not shift the d-band centre by almost 0.2eV, especially if the dopant does not even have d electrons ...

19) Could the authors please show that their results were converged at 400eV?

20) Which vdW correction did the authors choose?

21) No solvation corrections were mentioned. These are absolutely crucial for evaluating the selectivity of the CO₂RR. This needs to be fixed.

Remarks:

1) "Methanol Faradaic efficiency (FE) reported is lower than 50 % as the current density is higher than 50 mA/cm² (Table S1)." - "as" should be replaced with "when"

2) "It is no doubt that design of robust electrocatalyst for CO₂-to-methanol is highly desired." – "It" should be replaced with "There".

3) It would be nice to include some words like "Consequently" or "Furthermore" to link the sentences in the Introduction, but this is optional.

4) "In this approach, we use cation (Ag, Au, Zn, Cd) and anion (S, Se, I) to study" – missing word "doping"

5) "Using this universal method, Cu₂O/Cu matrix and other doping Cu₂O/Cu materials from mono to dual doping were also prepared for comparison" – Please rewrite this sentence, in its current form it makes no sense.

6) "including non-doping Cu₂O/Cu, mono doping S (or Ag)-Cu₂O/Cu," – Replace non-doping with non-doped, mono doping with mono doped.

7) increased from 2 umol 278 to 6 umol, - Please use the proper μ , throughout the text.

8) "Modeling structure showed that Cu₂O/Cu matrix was considered as loading of the Cu cluster on Cu₂O (111) surface bonding with the oxygen atoms" – Please rewrite this sentence.

9) "and shows the basically general trend" – please delete "basically"

10) "Ag,S-Cu₂O/Cu process more electrons" – "possess"?

11) "in banding *CO" – bonding

12) "It is obvious that the reactive activity of HER was dimmed with doping Ag" – please rewrite.

13) "gas absorption." – Unless the species were aBSorbed into the surface, this should be "aDsrption"

Reviewer #2 (Remarks to the Author):

The authors modify Cu₂O/Cu with silver and sulfur doping and they use it as electrocatalyst for CO₂ reduction to methanol in water/IL electrolyte. I believe the manuscript can be accepted in Nature Communications, but the manuscript can be improved after the following revision:

- Catalyst components can be determined in the electrolyte. Now the authors compare only atomic ratios.
- The energy diagrams consider only thermodynamics but the authors talk about "rate" determining steps. This is not correct.
- How does the CO₂ solubility depend on the IL/water mixtures?
- How does the Ag,S-doped catalyst perform in pure H₂O?
- The EIS was performed around the ocp. How different was the ocp for the catalysts tested? It is not obvious from the manuscript that all EIS analysis is relevant only for the ocp, which is surely far from the potential where the CO₂ reduction to methanol happens. Also, the charge transfer resistance looks too low.

Responses to the comments and revisions made

Reviewer #1 (Remarks to the Author):

In the present manuscript entitled “In situ dual doping for constructing efficient CO₂-to-methanol electrocatalysts”, the authors describe an investigation into a new oxide-derived copper structure utilizing dopant atoms to improve the CO₂RR activity and selectivity towards methanol. The authors find experimentally that Ag/S doped OD copper shows a higher selectivity towards methanol compared to other, previous approaches. They try to rationalize this with DFT calculations on somewhat related surface structures. In the following, I will mostly focus on the DFT modelling of the system, since this coincides with my experience. The experimental results appear to be reasonably well executed and described; however, I would leave that evaluation to an experimental electrochemist. Lastly, I found the choice of the “SCE” as a (reported) reference electrode, instead of the standard hydrogen electrode rather counterproductive, since it makes the comparison to other results and the computational treatment actively more complicated. Most of the manuscript is well written and argued, however, the computational part has several severe issues that the author s would have to address before any publication could even be considered. This includes the fact that the reported Gibbs adsorption energies in the SI do not reflect the displayed values in Figure 4b) which makes the conclusions of the paper questionable, at best. Lastly, large parts of the paper become unintelligible without constantly looking at SI figures, since the authors often do not even summarize the findings of the figures in the SI, which is against the idea of an SI.

I would either suggest to reject and resubmit or to perform major revisions to address the points below.

Response : We thank the reviewer for the very instructive comments. We have addressed all the concerns, as can be known from the responses to the following detailed comments.

In the following I will detail larger comments and smaller remarks (mostly spelling, grammar etc):

Comments:

Comment 1: In the introduction the authors state “and its production results in large amount of carbon emission globally.” – Could the authors please provide a citation for this, ideally including an estimation of the magnitude of these emissions?

Response 1: We thank the reviewer for the comment. In the introduction part of the revised manuscript, according to the comment, we have discussed this by “and the traditional process emits large amount of CO₂ (about 2.6 ton CO₂/ton methanol).⁷”. The related reference has also been cited (Ref. 7). Please see Page 2 in the revised manuscript.

Comment 2: The authors state “Methanol Faradaic efficiency (FE) reported is lower than 50 % as the current density is higher than 50 mA/cm² (Table S1).” – My initial request would have been to put the current number into context with other publications, however, table S1 already does that. It would be good to state this and to cite the publications that were used to generate table S1.

Response 2: We thank the reviewer for the comment. As suggested by the reviewer, in the revised manuscript, we have stated and cited the publications by adding “Literature survey shows that methanol could be produced on isolated Cu decorated carbon nanofibers with Faradaic efficiency (FE) of 44% and current density of 93 mA cm⁻².⁹ Boron phosphide exhibited methanol selectivity up to 92%, but the current density was only 0.2 mA cm⁻².¹⁰ Pd-Cu aerogel have also been employed as electrocatalyst for electrochemical synthesis of methanol with FE of 80% and current density of 31.8 mA cm⁻².¹¹ Besides, Cu selenide catalyst could boost CO₂ reduction to methanol with a FE of 77.6% and current density of 41.5 mA cm⁻².¹² Generally, methanol FE reported is lower than 50 % when the current density is higher than 50 mA cm⁻² (please see the details in Supplementary Table 1).^{5, 11-15}” Please see Page 2

in the revised manuscript.

Comment 3: It would be good to explain what the authors mean with the word “matrix”, since I would have expected the expression “host” in its stead. I am not against them using the word, however, please define it when it first appears.

Response 3: We thank the reviewer for the comment. We have replaced “matrix” with “host” in the revised manuscript.

Comment 4: Could the authors please provide their potentials vs standard hydrogen electrode, as is the de-facto standard in electrochemistry, rather than the obscure “saturated calomel electrode”.

Response 4: We thank the reviewer for the comment. The potentials were converted to the reversible hydrogen electrode (RHE) reference scale in the revised manuscript.

In the revised manuscript, according to the comment, we have emphasized this by “The potentials were converted to the reversible hydrogen electrode (RHE) reference scale using the relation $E_{\text{RHE}} = E_{\text{SCE}} + 0.244 + 0.059 \times \text{pH}$.”. Please see Page 21 in the revised manuscript.

Comment 5: There is little to no mention of DFT and its state of the art in the introduction.

Response 5: We thank the reviewer for the comment. According to the comment, in the revised manuscript, we have modified and added the discussion by “The density functional theory (DFT) calculations demonstrate that the anion S regulates the electronic structure of the adjacent Cu atom facilitating the formation of *CHO from *CO and the cation Ag mainly increases the reaction barrier of HER. Their synergistic interactions with the host material enhance the CO₂RR to methanol.” Please see Page 4 in the revised manuscript.

Comment 6: The Figures and Tables in the SI should be ordered in the order they appear in the main text. Alternatively, please provide a table of contents with links to each table and figure for easy access.

Response 6: We thank the reviewer for the comment. For easy access, a table of contents with links to each figure and table was added in the revised supplementary information.

Comment 7: The resolution of all figures is very low in the version that I reviewed. This can happen during the PDF generation process for the reviewers, but I would ask the authors to please check if they can provide the subfigures at a higher resolution or, ideally, provide them as vector graphics.

Response 7: We thank the reviewer for the comment. High resolution figures have been provided in this revision and the figures have been submitted to the journal in a separate high-resolution format.

Comment 8: Could the authors please comment on why they only tested the reference Cu₂O/Cu system for around 10 hours and all the others for up to 25 hours (Figure 2e)?

Response 8: We thank the reviewer for the comment. We tested the reference Cu₂O/Cu system for around 10 hours, because the FE of methanol decay rapidly over Cu₂O/Cu, approaching 0% after 10 hours. In the revised manuscript, we have emphasized this by “while the Cu₂O/Cu catalyst almost lost its catalytic ability toward methanol, approaching 0% after 10 hours.”. Please see Page 10 in the revised manuscript.

Comment 9: Could the authors please explain what the units and calculation method is for the data shown in Figure S14? It is certainly not Ag/Cu = 0.5. Also, the amount of Ag quickly drops by 75% and the amount of S drops by 90% ... I am not sure I would call that immediately "stable", however it seems

to stabilize long-term.

Response 9: We thank the reviewer for the comment. The data in supplementary Figure 14 is the atomic ratio of Ag/Cu and S/Cu during the process of Ag-Cu₂S to Ag,S-Cu₂O/Cu conversion, which was determined from the corresponding peak area ratio in the quasi-in-situ XPS spectra, and is dimensionless quantity (no unit). During this in situ reduction process, the precursor Ag-Cu₂S converted to Ag,S-Cu₂O/Cu, the atomic ratio of Ag/Cu and S/Cu decreased and remained constant after 10 min, which reveals that a stable Ag,S-Cu₂O/Cu was obtained after 10 min. This is consistent with the XRD and Raman results (Supplementary Figures 9 and 10). In supplementary Figure 14, we have added this by “It was calculated from the corresponding peak area ratio in the XPS spectra.” Please see Page 17 in the supplementary information.

Comment 10: The authors state “more active sites than nanoparticles for catalyzing the CO₂RR to methanol.” – Could the authors please provide data to back this up? E.g. EC surface area?

Response 10: We thank the reviewer for the comment. The results were provided in Supplementary Figure 25. Please see Page 28 in the supplementary information.

In the revised manuscript, according to the comment, we have also discussed this by “According to Supplementary Figure 25, the C_{dl} values of Ag,S-Cu₂O/Cu with nanonetwork structure (0.35 g CTAB) was 37.2 mF cm⁻², which was larger than that of Ag,S-Cu₂O/Cu with nanoparticle structure (0 g CTAB, 30.1 mF cm⁻²), Therefore, we can know that the Ag,S-Cu₂O/Cu with nanonetwork structure (0.35 g CTAB) had a larger ECSA. The porous nanonetworks resulted in more active sites than nanoparticles for catalyzing the CO₂RR to methanol.” Please see Page 11,12 in the revised manuscript.

Comment 11: Given that the authors generate an Ag/Cu system: Could the

authors please comment on how they make sure that they do not have an Ag/Cu tandem catalyst on the atomic level?

Response 11: We thank the reviewer for the comment. We agree with the reviewer that many bimetallic materials can act as tandem catalysts for the enhancement of CO₂RR product selectivity. For an Ag/Cu system, the improving selectivity may be mainly originated from Ag that can efficiently generate CO for further reduction (J. Am. Chem. Soc. 2019, 141, 2490–2499; Joule, 2020, 8, 1688-1699). However, the Ag and Cu in our catalyst do not constitute a tandem catalyst, as confirmed by our experimental findings. (i) The FE of methanol over Cu₂O/Cu, mono-doped Ag-Cu₂O/Cu and S-Cu₂O/Cu were 3.5 %, 25.0 % and 46.4 %, respectively (Fig. 2c). It indicates that both Ag-Cu and S-Cu system can promote methanol production. When the Ag and S are doped simultaneously, the FE of methanol is maximized (67.4 %), which should be attributed to the synergy of Ag and S, not just Ag. (ii) To further demonstrate this point, we design some control experiments. We performed the electrochemical test with CO as the feeding gas, assuming that enough CO was generating on the catalyst surface. The electrochemical performance was showed in Figure R1. It is obvious that the FE of methanol in the presence of CO is significantly lower than that of CO₂ as the feed gas. This suggests that sufficient CO is not a decisive factor for improving methanol selectivity over our Ag,S-Cu₂O/Cu catalyst. Based on the above results, and knowledge in the literature, we proposed that Ag and S can synergistically tune the electronic structure of nearby Cu active sites, making methanol production more accessible. This phenomenon has also been reported on other non-tandem Ag-doping catalysts, such as, the compressively strained CuAg surface alloys (J. Am. Chem. Soc. 2017, 139, 15848-15857) and the electro-deficient Cu₃Ag₁ catalyst (Adv. Energy Mater. 2020, 10, 2001987) etc.

Figure R1. Electrochemical performance of Ag,S-Cu₂O/Cu using CO₂ or CO as the feeding gas at the potential of -1.18 V vs. RHE.

Comment 12: While I know that oxygen atoms can be present in OD copper under normal CO₂RR conditions (~1-1.5V vs RHE), could the authors please find some justification why this is should still be the case at almost 2V vs RHE or more.

Response 12: We thank the reviewer for the comment. When we use the reversible hydrogen electrode (RHE) reference scale, the potential for the highest methanol FE (67.4%) is -1.18 V vs. RHE, which is among the normal CO₂RR conditions. Therefore, we think the presence of oxygen atoms in our catalysts is reasonable, as can be known from the XRD and XPS analysis (Supplementary Figures 9-13).

Comment 13: The resolution of the adsorbate structures in the SI figures is significantly too low to understand the adsorption structure.

Response 13: We thank the reviewer for the comment. The enlarged figures have been provided in the revised supplementary information. Please see Supplementary Figures 30-33 in the revised supplementary information.

Comment 14: The adsorption structures that were used in the DFT simulations need to be provided as part of the SI.

Response 14: We thank the reviewer for the comment. The adsorption structures have been provided in the revised supplementary information. Please see Supplementary Figures 30-33 in the revised supplementary information.

Comment 15: The authors make no note of how they chose these, very specific adsorption structures. Given that the surfaces they chose are highly undercoordinated a significant justification for their choice, over e.g. doped Cu(111) would be needed.

Response 15: We thank the reviewer again for the comment. Here we would like to discuss this briefly in response to the comment. Based on the XRD, HRTEM and XPS analysis, we are quite sure that the catalysts obtained by the in situ electrochemical conversion possess the interface structure of Cu₂O/Cu with different doping. Therefore, we theoretically simulated this Cu₂O/Cu interface structure host without considering the Cu(111). According to other experimental results, we demonstrated the doping effect over the Cu₂O/Cu interface structure on methanol product intermediates. We calculated several structures with Cu cluster supported on the Cu₂O, after optimization we obtained the most stable interface structure (Supplementary Figure 29). The adsorption structures are considered on these stable interface structures. Due to the presence of highly unsaturated coordination atoms at the interface, we chose them as the potential active sites. As the adsorption intermediates have a certain spatial structure, resulting in fewer sites that can satisfy the adsorption conditions. We compared different adsorption sites and finally selected the most stable adsorption intermediate structure with the lowest energy, as shown in Supplementary Figures 30-33.

As suggested by the reviewer, in the revised manuscript, we have made note in the section of Theoretical calculation and discussed this by “According to

our experimental results, we demonstrated the doping effect over the Cu₂O/Cu interface structure on methanol product intermediates. We calculated several structures with Cu cluster supported on the Cu₂O, after optimization we obtained the most stable interface structure (Supplementary Figure 29). The adsorption structures are considered on these stable interface structures. Due to the presence of highly unsaturated coordination atoms at the interface, we chose them as the potential active sites. As the adsorption intermediates have a certain spatial structure, resulting in fewer sites that can satisfy the adsorption conditions. We compared different adsorption sites and finally selected the most stable adsorption intermediate structure with the lowest energy, as shown in Supplementary Figures 30-33.”. Please see Page 22 in the revised manuscript.

Comment 16: Could the authors please provide a table with the charges assigned to each atom to showcase more clearly how the local charge distribution changes?

Response 16: We thank the reviewer for the comment. According to the reviewer’s comments, the Bader charge changes of each atom before and after doping were summarized in Supplementary Table 4, and the charge state changes around the dopant atoms were obvious. Please see Page 46-48 in the revised Supplementary information.

According to the comment, in the revised manuscript, we have discussed this “The Bader charge changes of each atom before and after doping were summarized in Supplementary Table 4, and the charge state changes around the heteroatoms were obvious. We also fitted a linear relationship between the oxidation state of the Cu atom adjacent to the dopant and the Bader charge in Supplementary Figure 37.” Please see Page 16 in the revised manuscript.

Comment 17: The authors claim that the “strong binding energy of *COOH onto the Cu₂O/Cu interface (-2.87 eV) leded[SIC] to the severe aggregation

of *COOH intermediates at the active sites and blocked the subsequent reaction path.” This is not in agreement with previous studies and the adsorption energy also does not coincide with the number the authors report in the SI, table S2 ($G(T)$, COOH, Cu₂O/Cu = 0.504eV), especially when viewed against the reported values for other surfaces ($G(T)$, COOH, Ag/S/Cu₂O/Cu = 0.502eV)

Response 17: We thank the reviewer for the comment. Fig. 4b is the reaction free energy (ΔG) for the corresponding intermediate adsorption reaction, which was calculated by $\Delta G_{(*\text{COOH})} = G_{(*\text{COOH})} - E_{(*)} - G_{(\text{CO}_2)} + 1/2G_{(\text{H}_2)}$. The $G(T)$ in Supplementary Table is different from ΔG , which was calculated by $G(T) = \text{ZPE} - \text{TS} + \text{H}(T)$. Therefore, the two values are different.

According to the comment, in order to avoid confusing, we have modified the ordinate labels of Fig. 4b and 4c in the revised manuscript. Moreover, we have replaced “strong binding energy” with “strong adsorption” in the revised manuscript. In the method section, we have also modified the description by “The free energy was obtained from $G = E + \text{ZPE} - \text{TS} + \text{H}$, where E is the total energy, H is the enthalpy, S is the entropy and ZPE is the zero-point energy at room temperature ($T = 298 \text{ K}$). The detailed values are displayed in Supplementary Table 3. The $G(T)$ in Supplementary Table 3 represents $\text{ZPE} - \text{TS} + \text{H}(T)$.” Please see Page 14 and 22 in the revised manuscript.

Comment 18: The authors claim “Taking into account that the Cu₂O/Cu has a much higher d-band center (relative to the Fermi level, -1.88 eV) compared with the doping catalysts (Figure S31),” – Firstly, this is, again, not readable without the SI, since the authors do not provide the comparison figure and leave the reader to go to the SI to look it up. Also, the introduction of 1-2 dopant atoms into a structure should not shift the d-band centre by almost 0.2eV, especially if the dopant does not even have d electrons.

Response 18: We thank the reviewer for the comment. We divide the comment into two sub-comments and answer them separately.

- (i) Firstly, this is, again, not readable without the SI, since the authors do not provide the comparison figure and leave the reader to go to the SI to look it up.

Response : According to the comment, in the revised manuscript, we have added the table of contents in the revised supplementary information for the convenience of readers.

- (ii) Also, the introduction of 1-2 dopant atoms into a structure should not shift the d-band center by almost 0.2eV, especially if the dopant does not even have d electrons.

Response : After reading the comment, we have read more papers in order to discuss the shift of d-band center with different dopants. The introduction of one or two dopant atoms seems to be faint, however the heteroatoms can make host structural changes, especially in the interface structures, which are also responsible for the shift in the d-band center (Nat. Commun. 2021,12, 1449; Angew. Chem. Int. Ed. 2021, 60, 22722-22728).

To make this discussion more readable, we have rewritten this sentence. "The projected density of state of the Cu in Cu₂O/Cu, Ag-Cu₂O/Cu, S-Cu₂O/Cu and Ag,S-Cu₂O/Cu is showed in Supplementary Figure 35. The Ag-Cu₂O/Cu (-2.16 eV, relative to the Fermi level), S-Cu₂O/Cu (-2.15 eV) and Ag,S-Cu₂O/Cu (-2.14 eV) catalysts have a lower d-band center than that of the Cu₂O/Cu (-1.88 eV) due to the doping effect and the local structural deformation.^{44,45}"

Please see Page 16 in the revised manuscript.

Comment 19: Could the authors please show that their results were converged at 400eV?

Response 19: We thank the reviewer for the comment. During structure optimization, the energy change criterion is set to 10⁻⁴ eV in the iterative solution of the Kohn-Sham equation. All the structures are relaxed until the

residual forces on the atoms have declined to less than 0.03 eV \AA^{-1} . When these criteria are reached, we consider the result to be convergent. Please see the section of Theoretical calculations in Page 22 of the revised manuscript.

Comment 20: Which vdW correction did the authors choose?

Response 20: We thank the reviewer for the comment. We have added this by “The long-range van der Waals interaction is described by the DFT-D3 approach.⁴⁸” Please see Page 22 in the revised manuscript.

Comment 21: No solvation corrections were mentioned. These are absolutely crucial for evaluating the selectivity of the CO₂RR. This needs to be fixed.

Response 21: We thank the reviewer for the comment. We agree with the reviewer that solvation corrections are crucial for evaluating the selectivity of CO₂RR. We have tried to fix it in our catalyst system, but it is difficult to achieve. Because when using implicit solvation correction, it is necessary to use a dielectric constant to simulate the solvent environment, but the electrolyte we use is an ionic liquid/water mixture solution, and the dielectric constant of this electrolyte is difficult to obtain. Moreover, the implicit solvation cannot show the real interaction between solvent and catalyst. The explicit solvation correction can avoid the above two issues. However, the explicit solvation correction is difficult to simulate the real experimental environment. In this study, we mainly focus on the effect of dual-doping atoms on the electrochemical CO₂RR performance of the catalysts. Therefore, we use the vacuum environment to conduct the theoretical simulation to investigate the property of catalysts, which is widely used currently (Nat. Commun. 2021,12, 2932; Nat. Commun. 2021, 12, 6022; Angew. Chem. Int. Ed. 2022, 61, e202112116). We believe that reliable theoretical calculation of solvent effect for such a complex reaction system is very challenging, and is very an interesting topic for further theoretical study.

Remarks:

1) "Methanol Faradaic efficiency (FE) reported is lower than 50 % as the current density is higher than 50 mA/cm² (Table S1)." - "as" should be replaced with "when"

Response: We have replaced "as" with "when" in the revised manuscript.

2) "It is no doubt that design of robust electrocatalyst for CO₂-to-methanol is highly desired." – "It" should be replaced with "There".

Response: We have replaced "It" with "There" in the revised manuscript.

3) It would be nice to include some words like "Consequently" or "Furthermore" to link the sentences in the Introduction, but this is optional.

Response: We have added the conjunction to link the sentences in the revised Introduction.

4) "In this approach, we use cation (Ag, Au, Zn, Cd) and anion (S, Se, I) to study" – missing word "doping"

Response: We have added the missing word "doping" in the revised manuscript.

5) "Using this universal method, Cu₂O/Cu matrix and other doping Cu₂O/Cu materials from mono to dual doping were also prepared for comparison" – Please rewrite this sentence, in its current form it makes no sense.

Response: This sentence was rewrote as following "To demonstrate this synthesis method is universal, other doping Cu₂O/Cu materials from mono to dual doping were also prepared (Supplementary Figures 3, 4 and 5)". Please see Page 5 in the revised manuscript.

6) "including non-doping Cu₂O/Cu, mono doping S (or Ag)-Cu₂O/Cu," – Replace non-doping with non-doped, mono doping with mono doped.

Response: We have replaced "non doping" with "non doped", "mono doping" with "mono doped" in the revised manuscript.

7) increased from 2 umol 278 to 6 umol, - Please use the proper μ , throughout the text.

Response: We thank the referee for the comment. The error has been corrected throughout the text.

8) “Modeling structure showed that Cu₂O/Cu matrix was considered as loading of the Cu cluster on Cu₂O (111) surface bonding with the oxygen atoms” – Please rewrite this sentence.

Response: This sentence was rewrote as following “Based on our previous results, we build the Cu₂O/Cu host structure by loading Cu cluster on the Cu₂O (111) surface, which is shown in Supplementary Figure 29a.”. Please see Page 13 in the revised manuscript.

9) “and shows the basically general trend” – please delete “basically”

Response : The “basically” has been deleted in the revised manuscript.

10) “Ag,S-Cu₂O/Cu process more electrons” – “possess”?

Response: We have modified the spelling error in the revised manuscript.

11) “in banding *CO” – bonding

Response: We have modified the spelling error in the revised manuscript.

12) “It is obvious that the reactive activity of HER was dimmed with doping Ag” – please rewrite.

Response: This sentence was rewrote as following “It is obvious that mono doping can increase the reaction barrier of HER (-0.23 eV and -0.34 eV for doping Ag or S, respectively, Fig. 4c) compared with non doped Cu₂O/Cu (-0.15 eV). In addition, the HER is further suppressed over the Ag and S dual-doped structure (-0.46 eV).”. Please see Page 16 in the revised manuscript.

13) “gas absorption.” – Unless the species were aBsorbed into the surface, this should be “aD sorption”

Response: We have replaced “absorption” with “adsorption” in the revised manuscript.

Reviewer #2 (Remarks to the Author):

The authors modify Cu₂O/Cu with silver and sulfur doping and they use it as electrocatalyst for CO₂ reduction to methanol in water/IL electrolyte. I believe the manuscript can be accepted in Nature Communications, but the manuscript can be improved after the following revision:

Response : We thank the reviewer for the very instructive comments. We have addressed all the concerns, as can be known from the answers to the following detailed comments.

Comment 1: Catalyst components can be determined in the electrolyte. Now the authors compare only atomic ratios.

Response 1: We thank the reviewer for the comment. The quasi-in-situ XPS and inductively coupled plasma (ICP) analysis were carried out to calculate the components of the catalyst in this revised manuscript. Please see Supplementary Figure 14 and Supplementary Table 2 in the revised manuscript.

According to the comment, in the revised manuscript, we have discussed this by “The quasi-in situ XPS revealed the existence of Ag and S species during the electrochemical reduction process, and the atomic ratios of Ag/Cu and S/Cu were unchanged after 10 min in situ conversion (Supplementary Figure 14). Moreover, the inductively coupled plasma (ICP) analysis verified the catalyst components. As shown in supplementary Table 2, the atomic contents of Cu, Ag and S in Ag,S-Cu₂O/Cu were 74.9%, 2.3% and 5.2%, respectively. Further estimations show that the molar ratio of metallic Cu and Cu₂O is about 1.1. The result is in consistent with the quasi-in situ XPS analysis.”. Please see Page 7 in the revised manuscript.

Comment 2: The energy diagrams consider only thermodynamics but the authors talk about "rate" determining steps. This is not correct.

Response 2: We thank the reviewer for the comment. In the revised

manuscript, according to the comment, we have modified the discussion by “Subsequently, the hydrogenation of *CO to *CHO is an endothermic process with the highest energy barrier in the methanol production process, representing the activity of catalyst for methanol production. The heteroatoms doping could effectively adjust the adsorption space position of the *CHO intermediate and its O atom (Fig. 4a). Over Ag,S-Cu₂O/Cu, the barrier energy for the hydrogenation of *CO to *CHO (0.66 eV) is lower than that over Ag-Cu₂O/Cu (0.88 eV) and S-Cu₂O/Cu (0.76 eV) (Fig. 4b)”. Please see Page 14 in the revised manuscript.

Comment 3: How does the CO₂ solubility depend on the IL/water mixtures?

Response 3: We thank the reviewer for the comment. After reading the comment, we have read more papers in order to discuss the relationship between CO₂ solubility and IL/water mixtures. The main advantage of using IL is that the strong absorption capacity of CO₂ can significantly increase the CO₂RR and inhibit the HER (Fluid Phase Equilibria, 2005, 228, 439-445.). It has been found that water has little effect on the physical adsorption capacities of CO₂ for ILs with non-basic anions, such as BF₄, PF₆ and NTf₂, which is due to the non-competition of water with the CO₂ sorption sites present in these ILs. (ChemSusChem, 2017, 10, 4927-4933; Journal of Chemical & Engineering Data, 2006, 51, 371-375; J Solution Chem 2013, 42, 1111-1122). Therefore, the CO₂ solubility in these ILs/water mixtures depends on the amount of ILs. Although the presence of a large proportion of IL can improve the adsorption of CO₂, the low mass transport rates also decrease CO₂RR performance due to the high viscosity of IL. Therefore, addition of certain amount of water to IL can lower the viscosity and tailor the proton concentration of the electrolyte, consequently affecting the CO₂RR rate (Science, 2011, 334, 643; Nat. Commun., 2014, 5, 4470; J. Electrochem. Soc., 2013, 160, H138). In this study, we found the current density increased to 122.7 mA cm⁻² with the highest FE of CH₃OH, when molar ratio of BMImBF₄/H₂O was 1:3. We have discussed the

effect of IL/water composition in Supplementary Figure 21. Please see Page 29 in the Supplementary information.

According to the comment, in the revised manuscript, we have discussed this by “IL can greatly improve the solubility of CO₂ in the electrolyte and ensure the effective supply of CO₂ during the CO₂RR process. However, the high viscosity of ILs would lead to lower mass transport and thus decrease the CO₂RR activity.^{39, 40}”. Please see Page 11 in the revised manuscript.

Comment 4: How does the Ag,S-doped catalyst perform in pure H₂O?

Response 4: We thank the reviewer for the comment. After reading the comment, we performed the CO₂RR electrolysis in pure H₂O electrolyte using Ag,S-doped catalyst. As shown in Supplementary Figure 21, the Ag,S-doped catalyst does not have the activity towards methanol with a very low current density of 0.51 mA cm⁻² in the pure H₂O. This should be caused by the poor conductivity of the electrolyte.

According to the comment, in the revised manuscript, we have discussed this by “It shows that the Ag,S-doped catalyst does not have the activity towards methanol with a very low current density of 0.51 mA cm⁻² in the pure H₂O electrolyte.”. Please see Page 24 in the revised Supplementary information.

Comment 5: The EIS was performed around the ocp. How different was the ocp for the catalysts tested? It is not obvious from the manuscript that all EIS analysis is relevant only for the ocp, which is surely far from the potential where the CO₂ reduction to methanol happens. Also, the charge transfer resistance looks too low.

Response 5: We thank the reviewer for the comment. As suggested by the reviewer, we carried out EIS experiment at the potential of -1.18 V vs. RHE where the CO₂ reduction to methanol happens. As shown in Supplementary Figure 26. The charges resistance (R_{ct}) on Ag,S-Cu₂O/Cu was much lower than that on other catalysts, which indicates a favorable kinetics on

Ag,S-Cu₂O/Cu towards CO₂RR. The small charge transfer resistance can be ascribed to the catalysts grow directly on the Cu foam substrate without binder, which would be beneficial for the enhancement of current density and catalytic activity.

According to the comment, in the revised manuscript, we have discussed this by “In addition, the electrochemical impedance spectrum (EIS, Supplementary Figure 26) was carried out to probe the effect of structural features on the charge transport kinetics at the potential of -1.18 V vs. RHE. It showed that charges resistance (R_{ct}) on Ag,S-Cu₂O/Cu was much lower than that on other catalysts, which indicate a favorable kinetics on Ag,S-Cu₂O/Cu towards CO₂RR.”. Please see Page 12 in the revised manuscript.

REVIEWERS' COMMENTS

Reviewer #1 (Remarks to the Author):

I thank the authors for addressing my concerns and remarks and clarifying some issues I had with the paper.

I had two remaining issues, which I think can be addressed with minor changes, after which the publication can proceed, in my opinion.

1) Following my earlier remark 5), it would be good to discuss some previous relevant publications in the field of electrochemical CO₂RR in order to give the reader some overview over previous findings.

2) It might be worthwhile to replicate the discussion surrounding the tandem catalyst in the SI, since some readers might have a similar concern.

Reviewer #1 (Remarks to the Author):

I thank the authors for addressing my concerns and remarks and clarifying some issues I had with the paper.

I had two remaining issues, which I think can be addressed with minor changes, after which the publication can proceed, in my opinion.

Comment 1: Following my earlier remark 5), it would be good to discuss some previous relevant publications in the field of electrochemical CO₂RR in order to give the reader some overview over previous findings.

Response 1: We thank the reviewer for the comment. As suggested by the reviewer, in the revised manuscript, we have added relevant discussion by “To date, there have been many efforts to achieve high methanol selectivity by constructing efficient electrocatalysts, such as metal alloys,⁹ metal chalcogenides,¹⁰ single-atom materials,¹¹ metal-organic compounds,¹² molecular catalysts¹³ and the pyridine-based catalysts¹⁴. Methanol could be produced on isolated Cu decorated carbon nanofibers with Faradaic efficiency (FE) of 44% and current density of 93 mA cm⁻².¹¹ Boron phosphide exhibited methanol selectivity up to 92%, but the current density was only 0.2 mA cm⁻².¹⁵ Pd-Cu aerogel have also been employed as electrocatalyst for electrochemical synthesis of methanol with FE of 80% and current density of 31.8 mA cm⁻².⁹ Besides, Cu selenide catalyst could boost CO₂ reduction to methanol with a FE of 77.6% and current density of 41.5 mA cm⁻².¹⁶ Generally, methanol FE reported is lower than 50 % when the current density is higher than 50 mA cm⁻² (please see the details in Supplementary Table 1).^{5, 9, 13, 16-18} Please see Page 2-3 in the revised manuscript.

Comment 2: It might be worthwhile to replicate the discussion surrounding the tandem catalyst in the SI, since some readers might have a similar concern.

Response 1: We thank the reviewer for the comment. As suggested by the reviewer, we have emphasized this by “Even though many bimetallic materials can act as tandem catalysts for the enhancement of CO₂RR product selectivity,

the Ag and Cu in our catalyst do not constitute a tandem catalyst, as confirmed by our experimental findings (Supplementary Fig. 30).” Please see Page 15 in the revised manuscript.

The relevant discussion was also added in the SI, please see Supplementary Fig. 30 in the revised Supplementary Information.